# Probing LLMs for Joint Encoding of Linguistic Categories

**Giulio Starace[★,1], Konstantinos Papakostas[1], Rochelle Choenni[1],**
**Apostolos Panagiotopoulos[1], Matteo Rosati[1], Alina Leidinger[1] and Ekaterina Shutova[1]**
[1]Institute for Logic, Language and Computation, University of Amsterdam

## Abstract

Large Language Models (LLMs) exhibit impressive performance on a range of NLP tasks, due to the general-purpose linguistic knowledge acquired during pretraining. Existing model interpretability research (Tenney et al., 2019) suggests that a linguistic hierarchy emerges in the LLM layers, with lower layers better suited to solving syntactic tasks and higher layers employed for semantic processing. Yet, little is known about how encodings of different linguistic phenomena interact within the models and to what extent processing of linguistically-related categories relies on the same, shared model representations. In this paper, we propose a framework for testing the joint encoding of linguistic categories in LLMs. Focusing on syntax, we find evidence of joint encoding both at the same (related part-of-speech (POS) classes) and different (POS classes and related syntactic dependency relations) levels of linguistic hierarchy. Our cross-lingual experiments show that the same patterns hold across languages in multilingual LLMs.

## 1 Introduction

Recent advancements in natural language processing (NLP) can be attributed to the development and pretraining of large language models (LLMs) such as BERT, GPT-3, and many others (Devlin et al., 2019; Brown et al., 2020; Touvron et al., 2023). For their intended use of providing general-purpose language representations suitable for many NLP tasks, these models must efficiently capture a wide range of linguistic features within their finite capacity. Despite their success, little is known about the way in which different types of linguistic information are organized in these models. Systematically understanding how these models represent linguistic phenomena and their interaction is crucial for the development of more effective NLP methods.

Existing research probed LLMs for their encoding of various linguistic properties such as agreement (Jawahar et al., 2019), word order and sentence structure (Tenney et al., 2018; Hewitt and Manning, 2019), co-reference (Tenney et al., 2019), semantics (Ettinger, 2020) and multilinguality (Ravishankar et al., 2019; Libovický et al., 2020). Taking a step further, Tenney et al. (2019) and Clark et al. (2019) studied where linguistic information is encoded in LLMs by probing different layers. Their results demonstrated that a linguistic hierarchy emerges in BERT representations, with lower layers capturing local syntax and higher layers being employed in higher-level semantic and discourse tasks. However, we do not yet understand how encodings of different linguistic phenomena interact within the models and to what extent processing of linguistically-related categories relies on the same, shared model representations.

There are many dependencies between processing different linguistic phenomena: for instance, information about a word's part of speech is likely to be employed when disambiguating its word sense. Alternatively, lower-level syntax is an important first step for semantic composition and natural language understanding tasks. In this work, we investigate how the (hierarchical) dependencies between different linguistic categories are encoded in LLMs, focusing on syntax. We ask a set of novel questions: (1) how related syntactic categories (e.g. different parts of speech (POS), such as *noun* or *verb*) are encoded within the models; (2) how syntactic categories at different levels of the linguistic hierarchy (e.g. POS classes and syntactic dependency relations) interact within the model; and (3) whether the observed patterns hold across languages.

To answer these questions, we propose a framework for testing the joint encoding of linguistic categories in LLMs. Specifically, we investigate to what extent the information about distinct linguistic categories is shared in the parameters of the

---

★ Corresponding author: giulio.starace@gmail.com.

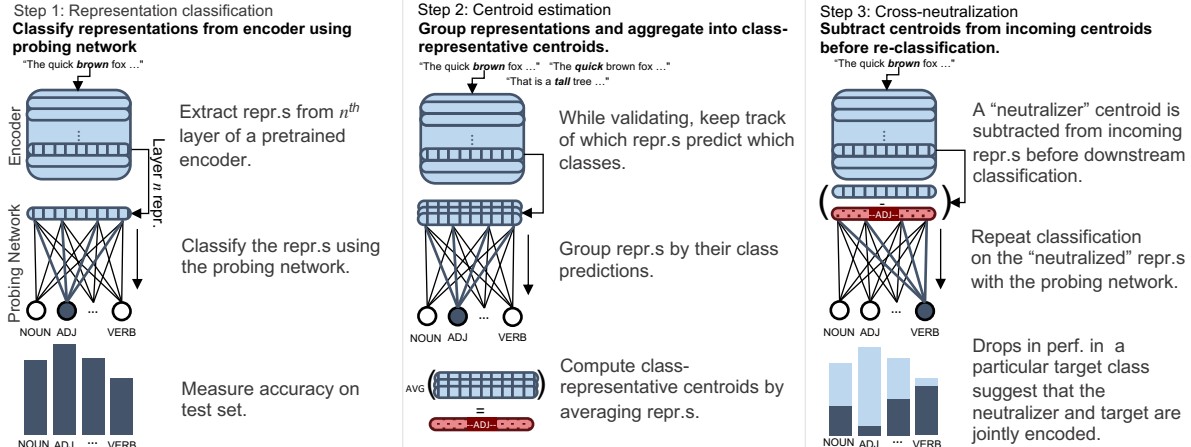

**Step 1: Representation classification**
**Classify representations from encoder using probing network**
"The quick **brown** fox …"

Extract repr.s from $n^{th}$ layer of a pretrained encoder.

Classify the repr.s using the probing network.

Measure accuracy on test set.

**Step 2: Centroid estimation**
**Group representations and aggregate into class-representative centroids.**
"The quick **brown** fox …"   "The **quick** brown fox …"
"That is a **tall** tree …"

While validating, keep track of which repr.s predict which classes.

Group repr.s by their class predictions.

Compute class-representative centroids by averaging repr.s.

**Step 3: Cross-neutralization**
**Subtract centroids from incoming centroids before re-classification.**
"The quick **brown** fox …"

A "neutralizer" centroid is subtracted from incoming repr.s before downstream classification.

Repeat classification on the "neutralized" repr.s with the probing network.

Drops in perf. in a particular target class suggest that the neutralizer and target are jointly encoded.

Figure 1: A diagram illustrating the three steps of our method: 1) Representation classification. 2) Centroid estimation. 3) Cross-neutralization. The complete methodology is outlined in Section 3.

model. Our approach (see Fig. 1) is inspired by the work of Choenni and Shutova (2022), who studied how LLMs share information across languages. We employ their *cross-neutralization* method, but extend it to study how information is encoded across linguistic categories in two syntactic tasks—part-of-speech (POS) tagging and dependency (DEP) labeling. In short, we test whether removing information on one syntactic category results in a failure to process another, related one. For instance, would removing a representation of the *verb* class hurt the model's ability to identify the verbs' syntactic dependencies, while still succeeding in this task on other categories, e.g. *nouns*? This provides insight into whether these categories are jointly encoded.

We study joint encoding patterns within both a monolingual and multilingual LLM, namely RoBERTa (Liu et al., 2019b) and XLM-R (Conneau et al., 2020). For the latter, we focus our analysis on English, Greek, and Italian, investigating to what extent multilingual models encode information for POS tags and DEP relations in a language-agnostic manner. We find that POS tags that are linguistically related are indeed jointly encoded by both the monolingual and multilingual models, and observe similar joint encoding patterns across all three languages. Moreover, we obtain further evidence of both language-agnostic and language-specific encoding within multilingual models, given that representations specific to POS tags and DEP relations can be approximately transferred across languages without a substantial impact on performance. Lastly, we find evidence of joint encoding between related POS tags and DEP relations, suggesting information sharing across

tasks at different levels of the linguistic hierarchy.

## 2 Background and related work

A wide range of methods have emerged to study the inner workings of neural networks (Belinkov and Glass, 2019; Madsen et al., 2022). Our approach is situated within the field of probing (Conneau et al., 2018), which typically involves the use of a simple auxiliary "probing" network trained to perform a specific task on the representations from a pretrained LLM. By keeping the probing network shallow and the pretrained model frozen, the predictions can be used to identify which information was already captured by the pretrained LLM.

**Previous work on probing**  Shi et al. (2016) are the first to introduce the idea of probing, training a logistic regression classifier on top of the embeddings from two neural machine translation encoders to study the syntactic information learned by them. Similarly, Adi et al. (2017) probe sentence representations for sentence length and word order by training auxiliary task classifiers on predicting these exact properties from the representations. Continuing on this work, Conneau et al. (2018) introduce a suite of probing tasks. Tenney et al. (2019) apply these to BERT's hidden states, quantifying where linguistic information is captured within the network. They find that the model follows the hierarchy of the traditional NLP pipeline, with lower-level, syntactic features appearing earlier than more complex semantic roles and structures. Ravishankar et al. (2019) bring the same analysis to the multilingual setting. Dalvi et al. (2021) find more supporting evidence of this

through an unsupervised approach. Manning et al. (2020) probe the attention mechanism in BERT for correspondence between linguistic phenomena such as syntactic dependencies and coreference. They also make use of structural probes (Hewitt and Manning, 2019), finding that the representations from BERT capture parse tree structure.

**Syntactic knowledge in BERT** Various approaches have been used to study what knowledge LLMs capture about syntax (Rogers et al., 2021). Previous probing studies have shown that BERT embeddings encode information about parts of speech, syntactic chunks, constituent and dependency labeling (Tenney et al., 2018; Liu et al., 2019a) as well as a broader set of syntactic features, such as tree depth and tense (Conneau et al., 2018). Yet, while it has become evident that syntax is captured to some extent, less is still known about where and how this information is acquired. Some works suggest that syntactic information is encoded in the token representations as they can be used to successfully reconstruct syntactic trees (Vilares et al., 2020; Kim et al., 2019; Hewitt and Manning, 2019). Others have instead studied syntactic knowledge at the level of attention heads, and show that particular heads specialize to specific aspects of syntax (Htut et al., 2019; Clark et al., 2019). However, Htut et al. (2019) find that these heads can not recover syntactic trees, suggesting that attention heads do not reflect the full extent of syntactic knowledge that these models learn.

**Information sharing in LLMs** Multiple works study information sharing in LLMs (Blevins et al., 2018; Şahin et al., 2020), with most focusing on cross-lingual sharing (Chi et al., 2020; Shapiro et al., 2021; Stanczak et al., 2022). Libovický et al. (2020) propose a simple method that removes language-specific information from model representations by capturing it through the mean of a set of representations from the respective language. Ravfogel et al. (2020) iteratively remove gender information in word embeddings through projection onto the nullspace (INLP) in order to mitigate bias in biography classification (De-Arteaga et al., 2019). Elazar et al. (2021) use INLP to build counterfactual representations for "amnesic probing". Here, the utility of a property for a given task is estimated by measuring the influence of removing the property via INLP, treating the removal as a causal intervention. Our work is inspired by Choenni and

|  | RoBERTa | XLM-R | | |
|---|---|---|---|---|
|  | en_gum | en_gum | it_vit | el_gdt |
| POS | 95.6% | 95.5% | 97.4% | 97.9% |
| DEP | 90.9% | 91.4% | 93.9% | 94.8% |

Table 1: Classification accuracy of our probing classifiers on English, Italian and Greek datasets for part-of-speech tagging and dependency labeling.

Shutova (2022), who probe for joint encoding of typological features across different languages. In particular, they probe LLMs for typological language properties and test whether subtracting language "centroids" from model representations negatively affects performance in typologically-related languages. While they focus on representations of specific languages, we target the representations of linguistic categories (e.g. nouns), and test whether these are jointly encoded across classes and tasks.

**POS tagging and dependency labeling** We study the joint encoding of POS categories and syntactic dependencies. Given a sentence, POS tagging is the task of mapping each word to the appropriate part of speech. For instance, in "The sailor dogs the hatch.", "dogs" is a verb, while in "He chases the dogs", "dogs" is a noun. Dependency labeling is the higher-level task of labeling the dependency relation between a "head" word and a "dependent" (or "parent" and "child"). For instance, in "That is a black car.", "black" is an *adjectival modifier* of "car".

## 3 Methodology

### 3.1 Probing

To study the representations from a given encoder network, we train a shallow classifier, or "probing classifier" on a probing task, using the representations from the encoder as input. By keeping the encoder frozen, we can ascribe all the learning to the relatively inexpressive probing classifier, allowing us to probe whether the representations from the encoder contain the information necessary to solve the task at hand.

### 3.1.1 Probing tasks

We focus on POS tagging and DEP labeling as our probing tasks. For both cases, our encoders take sentences as input, producing token-level embeddings at each layer. We extract the token-level embeddings from a given layer and pool them into word embeddings (detailed in Section 3.5). For

| Encoder / treebank | POS | | DEP | |
|---|---|---|---|---|
| | **Layer** | **Aggr.** | **Layer** | **Aggr.** |
| RoBERTa / en_gum | 3 | max | 3 | mean |
| XLM-R / en_gum | 9 | max | 9 | first |
| XLM-R / it_vit | 9 | first | 9 | mean |
| XLM-R / el_gdt | 12 | mean | 9 | mean |

Table 2: Optimal combination of embedding layer and subword pooling function for each encoder (RoBERTa & XLM-R), task (POS & DEP) and language (English, Italian & Greek) combination, chosen as outlined in Section 3.5 and used throughout our experiments. When neutralizing across languages (Section 5) and across tasks (Section 6), we use the configuration of the neutralizer for both neutralizer and target.

POS tagging, the probing classifier receives the word embeddings and is trained to classify them across 17 categories. For DEP labeling, we aim to label the dependency between child and parent. Thus, we pair each word in the sentence with its head as labeled in the dataset and concatenate their representations[1]. The probing classifier takes this concatenation as input and is trained to classify the dependency relation between the corresponding words across 36 categories.

### 3.1.2 Datasets

We use the Universal Dependencies treebanks (Nivre et al., 2020), manually annotated for POS tagging and DEP labeling. We choose the GUM (Zeldes, 2017), VIT (Delmonte et al., 2017), and the GDT (Prokopidis and Papageorgiou, 2017) datasets for English, Italian, and Greek respectively. All datasets contain word-level annotations with a total of 17 POS tags and 36 DEP relations shared across languages[2].

### 3.1.3 Models

**Encoders** We use RoBERTa (Liu et al., 2019b) and XLM-R (Conneau et al., 2020) as the encoders we probe. RoBERTa is an optimized version of the encoder-only transformer BERT (Devlin et al., 2019), and XLM-R is its multilingual variant, supporting 100 languages. Unlike BERT, both models are trained exclusively on the masked language modeling (MLM) objective. We use the "base" version of each model, comprising of 12 encoder layers, 12 attention heads and an embedding size of

768, with a total parameter count of $\sim$125 million[3].

**Probing classifier** For our probing classifier, we use a multilayer perceptron (MLP) consisting of two linear layers with a `tanh` activation function in between. We train our probing classifier using the AdamW optimizer (Loshchilov and Hutter, 2019) with a learning rate of $10^{-3}$ and weight decay of $10^{-2}$, and employ an early stopping criterion based on the validation set. We report the classification accuracy for the best configurations in Table 1.

### 3.2 Probing Classifier Selectivity Baseline

A sufficiently expressive probing classifier may be capable of learning *any* task given representations as input. This would render its probing functionality obsolete since the information localized by the probing classifier would no longer be necessarily attributed solely to the input representations. To ensure that our probing classifier is not overly expressive, we conduct a baseline check as outlined by Hewitt and Liang (2019). Here, we construct an analogous *control task* for POS tagging, where we randomly assign each word in the training set to one of $N$ arbitrary labels, where $N$ is the number of POS tag labels (i.e. 17). A good probe should have high *selectivity*, computed as the difference between the accuracy on the probing task and the accuracy on the control task. The more selective the probe, more likely it is that the information it accesses is specific to the input representations. If the probe performs equally well on the control task, it suggests that the probe may be leveraging some inherent properties of the model architecture or the data distribution, rather than specifically extracting useful information from the representations. We train a new checkpoint of our probing classifier on the control task, ending training after the same number of update steps performed when training on the POS tagging task, and report the outcome in Section 4.

### 3.3 Centroid estimation

To study the joint encoding of linguistic categories (e.g. nouns, verbs, etc.) in our encoders, we use the probing classifier to localize the subspace of the encoder that corresponds to each of these categories by obtaining their mean vector representation, or "centroid", similarly to the language centroids of

---

[1] Different methods such as adding the mean vector or absolute difference of the pair resulted in similar performance.

[2] We summarize the split sizes and an overview of the pre-processing pipeline for all three languages in Appendix A.

[3] We report the results of a brief parameter scaling experiment in Appendix B.

Libovický et al. (2020). The intuition is that representations that repeatedly result in predictions of a particular class will encode information specific to that class, which can be captured through aggregation such as averaging. For POS tagging, the centroid $u_t$ of each target POS class $t$ is defined as

$$\mathbf{u}_t^{(POS)} := \frac{1}{|V_t^l|} \sum_{\mathbf{v} \in V_t^l} \mathbf{v} \qquad (1)$$

where $V_t^l$ is the set of the representations of the words in our validation set that were predicted as tag $t$ when probing the $l$th layer of our the encoder. For DEP labeling, each prediction depends on both the representation of the head $\mathbf{h}$ and the child $\mathbf{c}$ of the DEP relation. The centroid of each target DEP relation $t$ is computed as:

$$\mathbf{u}_t^{(DEP)} := \frac{1}{|P_t^l|} \sum_{(\mathbf{h},\mathbf{d}) \in P_t^l} [\mathbf{h} \, ; \mathbf{d}] \qquad (2)$$

where $P_t^l$ is the set of (head, dependent) representation pairs that were predicted as DEP relation $t$ when probing the $l$-th layer of our encoder, and $[\mathbf{h} \, ; \mathbf{d}]$ is the concatenation of the representations.[4]

### 3.4 Cross-neutralization

To study whether different linguistic categories are jointly encoded within LLMs, we use a cross-neutralization method. We first evaluate the class-specific accuracy of our probing classifier on the original representations of our encoder. We then take a `neutralizer` centroid estimated as outlined in Section 3.3, and subtract it from the encoder representations corresponding to some `target` category. For instance, taking *verbs* as neutralizer and *nouns* as target, we subtract all *verb* information from the *noun* representations. We then repeat probing classification on these "neutralized" encoder representations. The intuition behind this is that if the encoders were to represent linguistic categories in independent ways, we expect the performance to deteriorate only for the linguistic category that we use for computing the centroids, i.e. removing *verb* information only negatively affects the representations of *verbs*. However, in the case of joint encoding, we expect to see substantial performance drops for other `target` categories as well,

---

[4]We test the quality of the predicted centroids by computing gold centroids based on the gold labels, and computing cosine similarity between the two. We find that they are near identical ($\sim$1 similarity), see Appendix C.

suggesting that some features of the encoder play a role in encoding both categories. As a baseline, we additionally experiment with subtracting random vectors (rather than centroids) from the encoder representations. If the centroids are indeed responsible for the drop in performance, then we should not observe similar performance drops when subtracting random vectors. We report the outcome of this experiment in Section 4.

### 3.5 Choosing a configuration for probing

Understanding where to best localize the task-specific information within the models is not trivial. For instance, while Tenney et al. (2019) show that syntax information is more localized in lower layers, de Vries et al. (2020) demonstrate that such findings do not automatically port to multilingual models. Moreover, Del and Fishel (2022) show that for multilingual models, the pooling method used can have important effects on the knowledge that is captured. Thus, we first study which layer $l$ and pooling function maximizes the amount of information that is captured in our centroids. As such, we employ the notion of *self-neutralization*, the case in which the same linguistic category is both the target and the neutralizer.

We hypothesize that a large drop in accuracy after self-neutralization is indicative of a high amount of relevant information being captured by the centroid. This drop should be relative to a high accuracy. We thus select the top quartile of our configurations in terms of original-encoder accuracy, and from this subset use the configuration with the highest relative drop due to self-neutralization. We consider representations from layers $l \in \{1, 3, 6, 9, 12\}$ and pool subwords by either taking the first token in a word, or by max-pooling or mean-pooling over the token representations. For each encoder, language and task, we separately find the optimal probing configuration and report them in Table 2. We make use of the validation set for this portion of the methodology.

## 4 Joint encoding of POS tags

We begin by examining whether representations of different POS tags share information. For each POS tag, we compute its centroid and re-classify representations neutralized by the centroid. With regard to our baselines, we find that subtracting random vectors leaves the performance relatively unchanged, with no evidence of systematic drops.

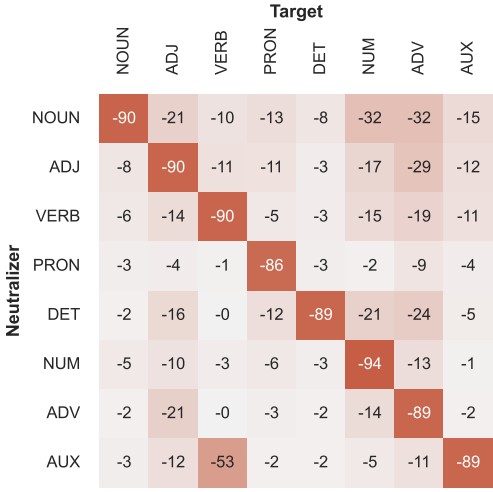

Figure 2: Relative change in accuracy when cross-neutralizing POS tags (RoBERTa).

For our selectivity baseline, we find that our probing classifier achieves an average accuracy of 96% on the POS tagging task and 63% on the control task, giving a selectivity value of 33%. This is in line with probes for similar tasks accepted for their validity in the literature (Hewitt and Liang, 2019).

To facilitate visualization and discussion, we select an illustrative subset[5] of POS tags that covers both open- and closed-class words, as we expect potentially different patterns between these two groups. For instance, we expect neutralization to be symmetric (neutralizer and target can be swapped for similar results) in the case of open-class words as we hypothesize that joint encoding here is dictated by co-occurrence patterns. Conversely, we expect asymmetric neutralization in the case of close-class neutralizers, supposing that joint encoding here may be dictated by functional dependencies, which may not necessarily be reciprocal.

**Results**   We find that linguistically related categories generally tend to be jointly encoded in RoBERTa (see Fig. 2). For instance, the relative decrease of VERB classification accuracy by 53% when neutralizing using auxiliaries (AUX) suggests information sharing. This may be explained by the fact that auxiliaries themselves are verbs (e.g. "*has done*"), and that they functionally modify verbs. We note that the information flow in this pairing is asymmetric, as noted by the much smaller 11% drop in the reciprocal case. This aligns with our

---

[5]See Appendix F for results between all classes.

hypothesis that joint encoding for closed-class neutralizers may be explained by their functional role, which is often not reciprocal (verbs will generally not modify auxiliaries).

While we observe further evidence of linguistically related categories being jointly encoded (e.g. VERB and ADV), this is not ubiquitous. For instance, joint encoding is not found when neutralizing nouns (NOUN) with determiners (DET). Here, the relative percentage decrease is only 2%, despite determiners acting as modifiers for nouns. In general, we do not find distinct patterns of joint encoding unique to closed-class and open-class words. Our hypothesis of symmetry due to co-occurrence is at best supported by the ADV-ADJ pairing, with relative drops of -21% and -29%. In most other pairings information sharing appears to be asymmetric, suggesting functional dependence or another root cause as explanation.

Moreover, we note that open-class words tend to be "adept" at cross-neutralizing, sharing information as neutralizers with many other tags as shown by the NOUN, ADJ and VERB neutralizers rows successfully neutralizing several columns. However, we note similar patterns when neutralizing with the closed-class DET. Overall, we find evidence of joint encoding of POS tags in RoBERTa, but further work is necessary to establish which mechanisms lead to specific pairs sharing information or not.

### 4.1   Joint encoding in multilingual models

We now investigate how our findings transfer to other languages. We examine to what extent multilingual models demonstrate language-specific or language-agnostic information-sharing behavior. Additionally, we verify the consistency of our method across models as we can directly compare the results between RoBERTa and XLM-R in English, and also across languages, as we can compare the effects of cross-neutralization on English, Italian, and Greek. Our setup is the same as in Section 4, replacing RoBERTa with XLM-R and repeating the experiment in the three languages.

**Monolingual vs. multilingual model**   When comparing RoBERTa and XLM-R on the English dataset, we observe similar patterns across POS tags (see Fig. 2 and the left-most column of Fig. 3). For instance, for both models, we see that nouns have a neutralization effect on adjectives, numerals, and adverbs. This suggests that given a language, multilingual models jointly encode POS tags simi-

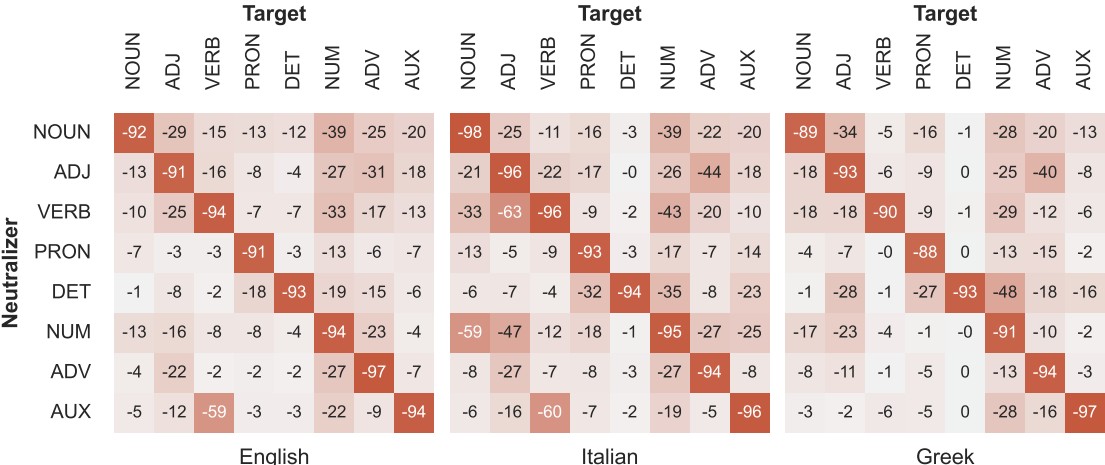

Figure 3: Relative change in accuracy when cross-neutralizing POS tags using embeddings from XLM-R in (left-to-right respectively) English, Italian and Greek, labeled by their universal dependencies treebank.

larly to their monolingual counterparts.

**Joint encoding within languages**  We find that XLM-R exhibits similar sharing patterns within different languages (see Fig. 3). For instance, we see that adjectives consistently neutralize adverbs across all languages (-31%, -44% and -40% in English, Italian and Greek). This suggests that the representations responsible for predicting these tags contain some language-agnostic information.

However, we also observe some language-specific behavior. For instance, VERB categories cross-neutralize adjectives (ADJ) more prominently in Italian (-63%) than in English (-26%) and Greek (-18%). This may be explained by the postnominal use of adjectives in Italian which may overlap more closely with the positioning of verbs, particularly in the past participle tense (Cinque, 2010) (in contrast to the strictly prenominal use in the other languages). The results suggest that XLM-R discovers and leverages language-agnostic information when possible, while also learning language-specific information when necessary.

## 5  Information sharing across languages

Our findings for individual languages on XLM-R raise the question of whether information about linguistic categories from two *different* languages can also be jointly encoded. To further probe for language-agnostic joint encoding of linguistic categories, we extend our experiment to test whether information between two linguistic categories from two different languages is jointly encoded. We posit that if categories share information across

languages, this is evidence that the models learn at least partially language-independent representations for these categories. We test this hypothesis by cross-neutralizing every linguistic category in a language $A$ with another category from a language $B$, e.g. neutralizing all Italian POS tags with English nouns. Aside from probing for language-agnosticism, this allows for further exploration of joint encoding of linguistic categories. To enable consistent neutralization, we use the same layer and pooling function configuration (see Section 3.5) for both neutralizer and target. We do this by utilizing the neutralizing language configuration for both neutralizer and target in cases where they differ.

**Results**  Fig. 5 shows results of cross-neutralizing for Italian POS tags when using the corresponding English centroids from each linguistic category. On the diagonal, neutralizer and targets correspond to the same linguistic category, but for different languages. Accuracy drops substantially here ($\sim 70\%$ on avg.). This is evidence of language-agnostic encoding of these linguistic categories, as centroids in one language neutralizing targets in another language points to information in the representations being shared across both languages.

We also observe joint encoding of different linguistic categories, for instance with English NOUNs neutralizing Italian numerals (NUM) (-48%), or English adjectives (ADJ) neutralizing Italian ad-

---

[6]Note that because the linguistic categories are now different between neutralizer and target, the diagonal no longer represents self-neutralization, and is simply an artifact of the ordering of the categories to favor legibility.

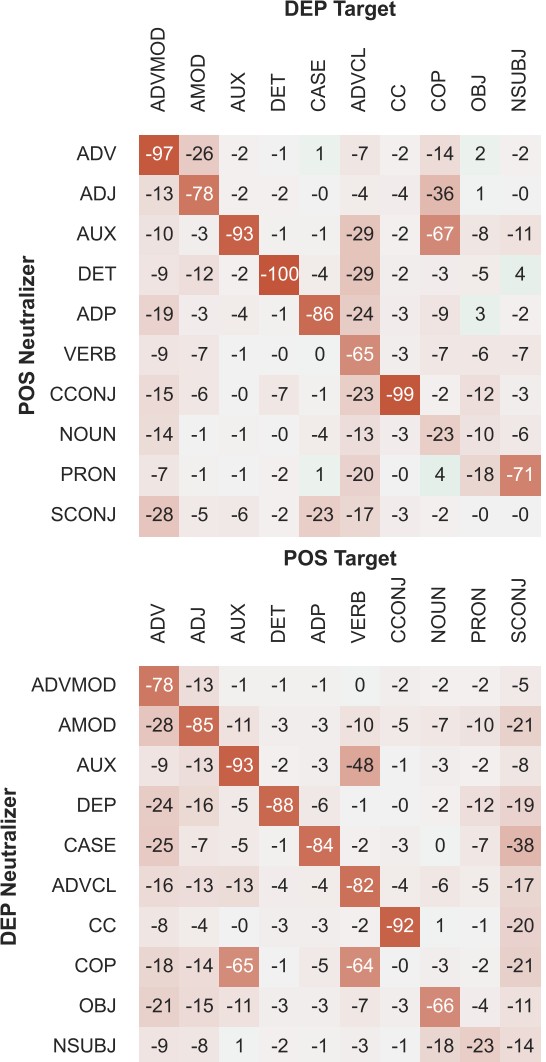

Figure 4: Relative change in accuracy cross-neutralizing RoBERTa DEP representations using RoBERTa POS centroids (top) and vice versa (bottom)[6].

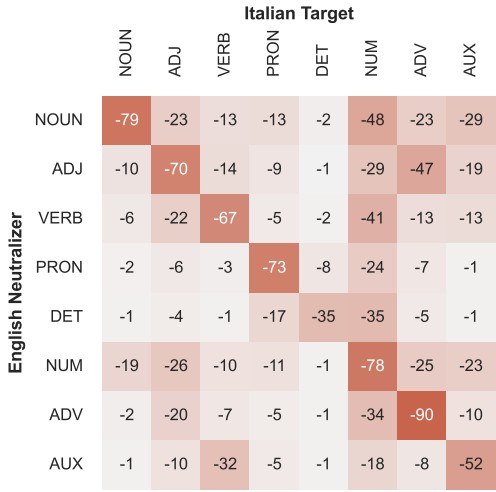

Figure 5: Relative change in accuracy when cross-neutralizing XLM-R embeddings in Italian using centroids from English.

verbs (ADV) (-47%). An alternative explanation to language-agnostic representation learning in XLM-R may be that instances of code-switching or language borrowing in the pretraining data may have encouraged XLM-R to jointly encode different linguistic categories across languages. Future work may investigate why joint encoding occurs across different categories of different languages.

We repeat the experiment using English neutralizers on the Greek corpus, and observe similar but milder trends. This may be due to English being phylogenetically closer to Italian than Greek (Chang et al., 2015), leading to a higher degree of information sharing between the former two languages, but further work in determining which language pairs share more information is needed

to draw solid conclusions. For more detailed plots on cross-lingual cross-neutralization, we refer the reader to Appendix F, where we present results from all combinations of neutralizer and target across the three languages.

## 6 Information sharing across tasks

To study whether information is shared across the linguistic hierarchy, we cross-neutralize between tasks using POS neutralizers and dependency relation targets. Since a given parent may have several dependents while the child will only have one parent, we posit that most of the information for the child-parent dependency is captured by the child. We therefore subtract the POS centroids from the child representations in the child-parent concatenations for dependency labeling. We also complete the complementary experiment of subtracting the child portion of the dependency relation centroids from the POS embeddings for POS tagging, to test whether joint encoding happens in both directions. As in Section 5, we use the neutralizer configuration for both neutralizer and target for consistency.

**Results** Fig. 4 (top) shows the results when neutralizing RoBERTa representations for DEP labels using POS centroids. We find further evidence of linguistically related units being jointly encoded. For instance, adverb POS tags (ADV) neutralize adverbial modifier (ADVMOD) dependency labels, as can be seen by the 97% relative drop in accuracy. The fact that POS tag representations jointly contain information that is crucial for encoding DEP

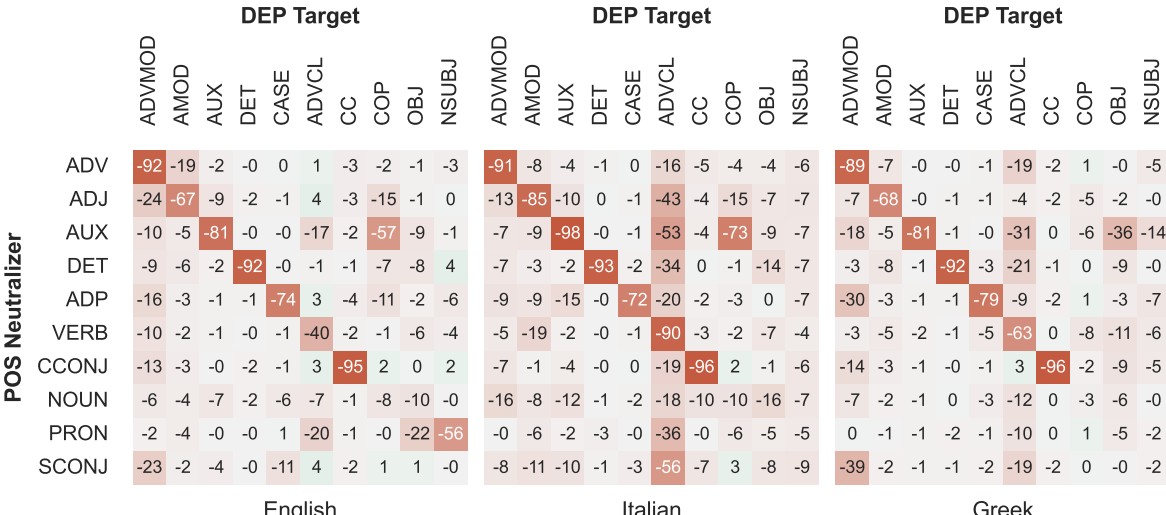

Figure 6: Relative change in accuracy for a sample of dependency relations when cross-neutralizing XLM-R DEP embeddings using XLM-R POS centroids in (left-to-right respectively) English, Italian and Greek.

labels shows that LLMs learn hierarchically, reminiscent of the classical NLP pipeline.

To study whether our findings generalize beyond English and RoBERTa, we repeat the experiment on XLM-R (see Fig. 6). Large drops in accuracy on the diagonal suggest that many information-sharing patterns hold across languages and models, consolidating our findings from sections 4.1 and 5. We also find language-specific results: pronouns (PRON) neutralize nominal subjects (NSUBJ) only in English (-56%), having surprisingly little effect on the other languages (-5% and -2%). On occasion, we find evidence of bidirectional sharing between different levels of the linguistic hierarchy. In Fig. 4 (bottom), we note that certain pairs highlighted by the diagonals such as adpositions (POS: ADP) and case relations (DEP: CASE) present evidence of joint encoding both when neutralizing DEP with POS centroids (-55%) and when neutralizing POS with DEP centroids (-84%). This seems to suggest that the hierarchical nature of the representations learned by these LLMs is not necessarily unidirectional: information appears to be shared both upwards and downwards in the linguistic hierarchy.

## 7 Conclusion

We study information sharing between linguistic categories in LLMs, finding evidence of joint encoding between pairs of related POS tag classes. By applying our method to XLM-R, we find evidence of joint encoding in XLM-R across languages, showing that our findings hold for both the monolingual (RoBERTa) and multilingual (XLM-R) case. Lastly, we cross-neutralize between POS tagging and DEP labeling, and find evidence of information sharing across the linguistic hierarchy. We test specifically for joint encoding of different syntactic categories that rely on the same linguistic concept, such as an "amod" dependency implying some relationship with the adjective and the noun POS tags. However, our method could be extended to test for joint encoding of categories in other tasks that can be expected to share information. This in turn could be informative on whether an LLM indeed captures this shared information between different tasks. A more complete map of this knowledge could help develop better models in many application scenarios: for instance, in a multitask learning setting where negative interference between parameter updates from different tasks is known to hamper performance (Zhao et al., 2018) or a lifelong learning setting where learning a new task often leads to (catastrophic) forgetting of the previously learned ones (Biesialska et al., 2020). Future work may additionally seek to explore more languages, particularly low-resource and typologically distant languages, or consider including higher-level semantic tasks from the classical NLP pipeline, such as semantic role labeling, word sense disambiguation and coreference resolution. Further work is also necessary for understanding *why* certain representations share information while others do not.

## Limitations

In this work, we limit our experiments to an encoder-only architecture. Further research could be carried out on encoder-decoder or decoder-only architectures.

Additionally, our experiments only examine three languages. Future work may therefore aim to extend the experiments to include more languages, particularly low-resource languages. Similarly, our discussion of the linguistic hierarchy is limited to two tasks from the early stages of the classical NLP pipeline. As mentioned, further work may extend the experiments to higher-level semantic tasks.

Lastly, the lack of cross-neutralization effects between some linguistic categories does not necessarily indicate the absence of joint encoding, but merely that our representations were not sufficient to prove its existence.

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

## A  Dataset pre-processing

Table 3 presents the train, validation and test split sizes (in terms of number of sentences) for each language considered in our work.

The sentences in the corpora from the Universal Dependencies framework are already tokenized to the word level and stored as lists of words in the `tokens` field. However, since we use sub-word tokenizers, namely the Byte Pair Encoding (Sennrich et al., 2016) and SentencePiece (Kudo and Richardson, 2018) tokenizer, we further split the words into their sub-word tokens. Depending on the task, we also include either the upos field, which is a list of integers corresponding to one of the 17 universal POS tags available, or the head and deprel fields which contain the head and one of the 36 dependence relations for dependency labeling[7]. It should be noted that we only keep the language-independent relations, as some of them appear only with a language-specific modifier, and including them would make comparisons across languages less straightforward[8].

Furthermore, upon inspecting the datasets we observed that the annotators had split contractions into their parts and included them next to the original contraction for the Italian and Greek corpora. However, ground-truth labels were only provided for the sub-words, with the compound words annotated as a special class "_". Hence, we filtered out the compound words from these datasets and retained their sub-parts. In addition to that, for dependency labeling, we ignored words with the root dependency label, as they have no head and their prediction is trivial.

## B  Scaling to larger models

We repeat the experiments from Sections 4 & 5 to verify whether they hold across different model sizes. More specifically, we scale the English POS tag cross-neutralization from RoBERTa-base to RoBERTa-large (Fig. 10a), and from XLM-R-base to XLM-R-large (Fig. 10b) and XLM-R-XL (Fig. 10c). We notice that similar patterns occur across all tested model sizes (e.g. NOUN neutralizing ADV). The effect appears to be generally less pronounced for larger models, although we also did find instances where the drop in accuracy is the same or higher.

---

[8]A full list of POS tags and dependency relations can be found on the Universal Dependencies website.

## C  Golden and Predicted Centroid Similarity

We present the cosine similarity between golden and predicted centroids for the POS tagging setup in table 4, as mentioned in Section 3.3.

## D  Self-neutralization visualization

Figs. 8 and 7 showcase the decrease in accuracy when self-neutralizing in the POS tagging/dependency labeling task for RoBERTa and XLM-R accordingly.

## E  Engineering Logistics

We rely on PyTorch Lightning (Falcon and The PyTorch Lightning team, 2019) and Hugging Face Datasets (Lhoest et al., 2021) and Transfomers (Wolf et al., 2020) for our implementation. We run our experiments on an NVIDIA A100 GPU, with each training run taking approximately three minutes. Our code is available at github.com/thesofakillers/infoshare.

## F  Detailed Cross-Neutralizing Results

We display the complete results for our cross-neutralization experiments in Figs. 9 – 18. More specifically, Fig. 9 corresponds to the monolingual setting with RoBERTa on English, and Fig. 11 to the multilingual setting with XLM-R on English, Italian and Greek. Figs. 12 – 14 show the cross-lingual setting for every possible pairing of our three languages with XLM-R. Finally Figs. 15 and 16 show the monolingual cross-task setting in RoBERTa in POS-DEP and DEP-POS directions while Figs. 17 and 18 show the same but in the multilingual setting with XLM-R.

| | Train | Validation | Test |
|---|---|---|---|
| **English (GUM)** | 4287 | 784 | 890 |
| **Italian (VIT)** | 8277 | 743 | 1067 |
| **Greek (GDT)** | 1662 | 403 | 456 |

Table 3: Sentence count in each split of each of the datasets we consider, averaging 24 words per sentence.

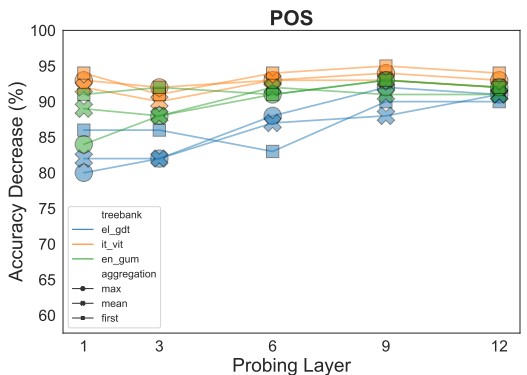
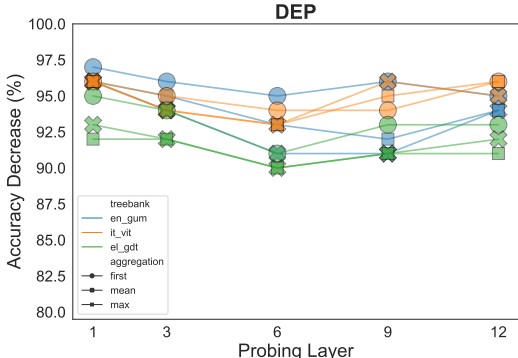

(a) POS tagging Accuracy Decrease using different WordPiece pooling methods ("aggregation" in the plot).

(b) Dependency labeling Accuracy Decrease using different WordPiece pooling methods ("aggregation" in the plot).

Figure 7: Decrease in performance for XLM-R when **self-neutralizing** in the POS tagging (left) and dependency labeling (right) tasks using embeddings extracted from different layers and setup configurations, for each of for English (en_gum), Italian (it_vit) and Greek (el_gdt) treebanks. For dependency labeling, we use the best child-head concatenation mode (ONLY) based on the results we aquired with RoBERTa, as shown in Fig. 8b.

Table 4: Cosine similarities between golden and predicted centroids for the RoBERTa POS tagging setup.

| POS Tag | Cosine Similarity |
|---------|-------------------|
| NOUN | 1.000 |
| ADP | 1.000 |
| ADJ | 1.000 |
| PUNCT | 1.000 |
| AUX | 1.000 |
| VERB | 1.000 |
| DET | 1.000 |
| CCONJ | 1.000 |
| NUM | 1.000 |
| ADV | 0.9999 |
| SCONJ | 0.9998 |
| PRON | 1.000 |
| PART | 1.000 |
| SYM | 0.9951 |
| PROPN | 0.9999 |
| X | 0.9737 |
| INTJ | 0.9980 |

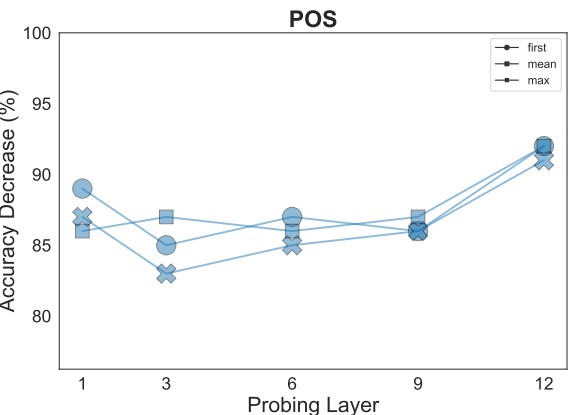

(a) Accuracy decrease for **POS tagging** when using different WordPiece poolings.

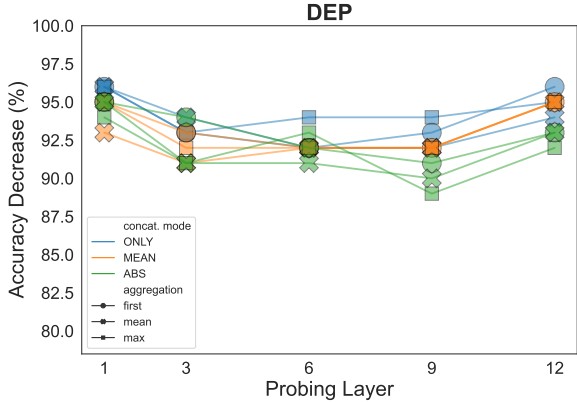

(b) Accuracy decrease for **dependency labeling** when using different WordPiece pooling methods ("aggregation" in the plot) and child-head concatenation configurations.

Figure 8: Decrease in performance for RoBERTa when **self-neutralizing** in the POS tagging (top) and dependency labeling (bottom) tasks using embeddings extracted from different layers and setup configurations.

**Target**

| Neutralizer | ADJ | ADP | ADV | AUX | CCONJ | DET | NOUN | NUM | PART | PRON | PUNCT | SCONJ | VERB | avg |
|---|---|---|---|---|---|---|---|---|---|---|---|---|---|---|
| ADJ | -90 | -4 | -29 | -12 | -6 | -3 | -8 | -17 | -11 | -11 | -0 | -18 | -11 | -10 |
| ADP | -7 | -89 | -24 | -8 | -3 | -2 | 0 | -23 | -27 | -8 | 0 | -39 | -1 | -8 |
| ADV | -21 | -2 | -89 | -2 | -3 | -2 | -2 | -14 | -5 | -3 | 0 | -6 | -0 | -4 |
| AUX | -12 | -4 | -11 | -89 | 0 | -2 | -3 | -5 | -9 | -2 | 0 | -11 | -53 | -9 |
| CCONJ | -4 | -3 | -8 | -0 | -94 | -3 | 1 | -2 | -2 | -1 | 0 | -21 | -2 | -3 |
| DET | -16 | -6 | -24 | -5 | -0 | -89 | -2 | -21 | -8 | -12 | 0 | -19 | -0 | -8 |
| NOUN | -21 | -4 | -32 | -15 | -8 | -8 | -90 | -32 | -22 | -13 | -0 | -30 | -10 | -22 |
| NUM | -10 | -1 | -13 | -1 | -0 | -3 | -5 | -94 | -4 | -6 | 0 | -4 | -3 | -5 |
| PART | -2 | -6 | -13 | -6 | -0 | 0 | 0 | 0 | -96 | -2 | 0 | -19 | -4 | -3 |
| PRON | -4 | -1 | -9 | -4 | -4 | -3 | -3 | -2 | -10 | -86 | -0 | -18 | -1 | -5 |
| PUNCT | -7 | -6 | -31 | -10 | -10 | -5 | -5 | -9 | -5 | -5 | -62 | -20 | -5 | -12 |
| SCONJ | -4 | -5 | -27 | -2 | -2 | -0 | 0 | -5 | -6 | -1 | -0 | -86 | -0 | -2 |
| VERB | -14 | -5 | -19 | -11 | -4 | -3 | -6 | -15 | -11 | -5 | 0 | -21 | -90 | -11 |

Figure 9: Relative change in accuracy when cross-neutralizing POS tags using centroids from RoBERTa in English.

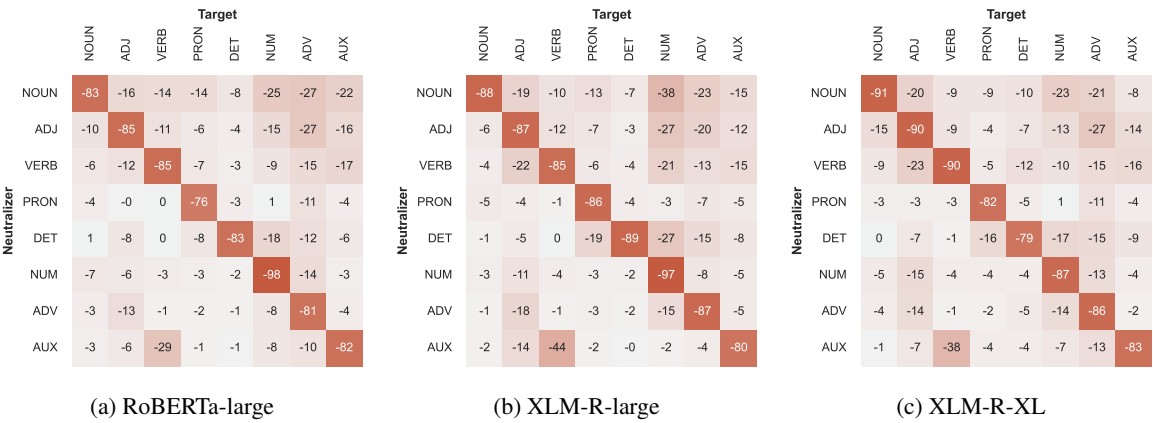

(a) RoBERTa-large     (b) XLM-R-large     (c) XLM-R-XL

Figure 10: Change in accuracy when cross-neutralizing English POS tags with centroids from different model sizes.

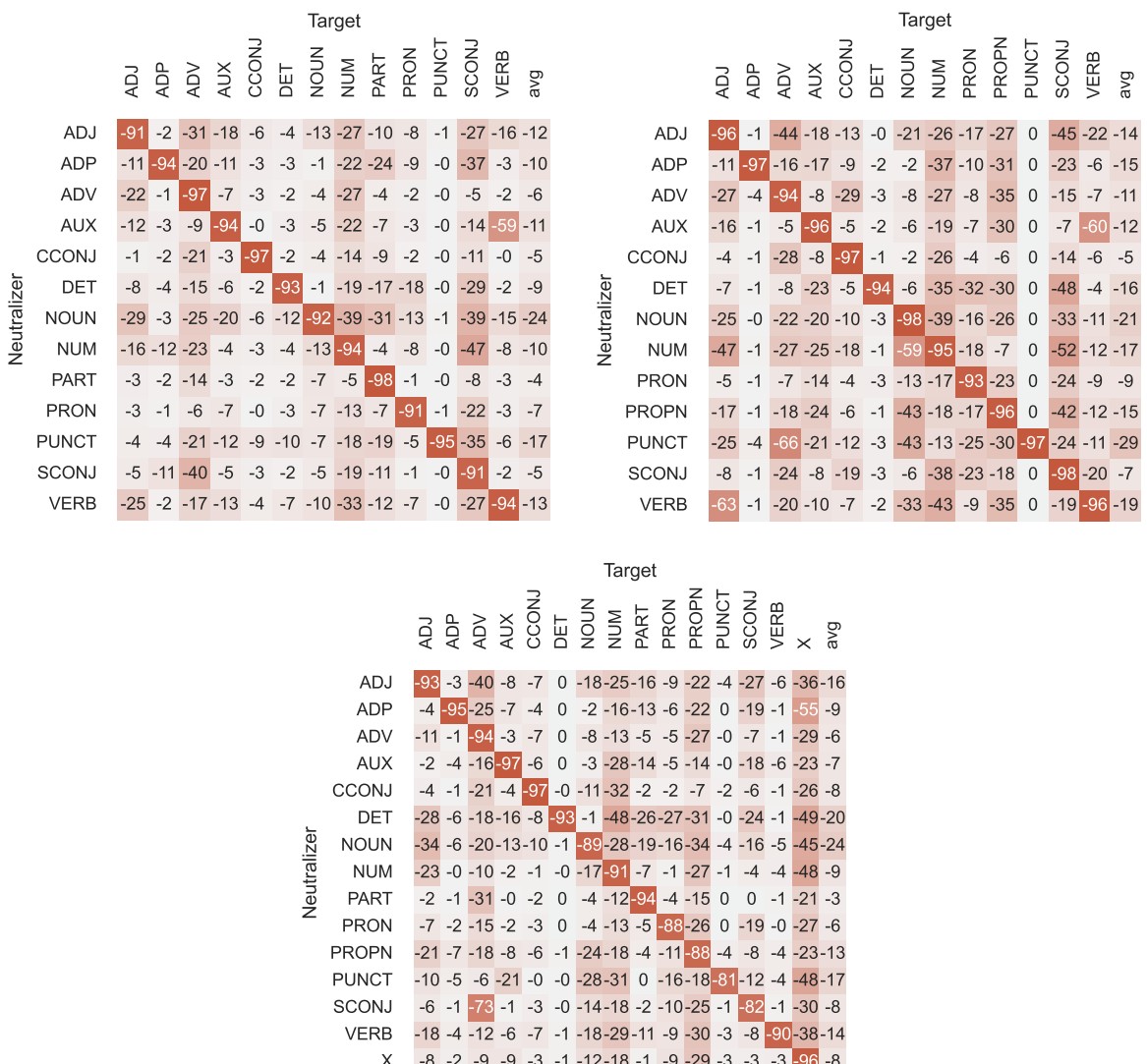

Figure 11: Relative change in accuracy when cross-neutralizing POS tags using centroids from XLM-R. In English (top, left), Italian (top right), and Greek (bottom).

**Italian Target — English Neutralizer (left)**

| | ADJ | ADP | ADV | AUX | CCONJ | DET | NOUN | NUM | PRON | PROPN | PUNCT | SCONJ | VERB | avg |
|---|---|---|---|---|---|---|---|---|---|---|---|---|---|---|
| ADJ | -70 | -1 | -47 | -19 | -9 | -1 | -10 | -29 | -9 | -12 | 0 | -34 | -14 | -10 |
| ADP | -8 | -63 | -12 | -13 | -5 | -2 | -0 | -44 | -12 | -22 | 0 | -27 | -5 | -10 |
| ADV | -20 | -1 | -90 | -10 | -12 | -1 | -2 | -34 | -5 | -19 | 0 | -10 | -7 | -7 |
| AUX | -10 | -0 | -8 | -52 | -5 | -1 | -1 | -18 | -5 | -8 | 0 | -10 | -32 | -6 |
| CCONJ | -2 | -1 | -23 | -6 | -82 | -1 | -2 | -27 | -6 | -2 | 0 | -11 | -7 | -4 |
| DET | -4 | -1 | -5 | -1 | -3 | -35 | -1 | -35 | -17 | -12 | 0 | -37 | -1 | -6 |
| NOUN | -23 | -0 | -23 | -29 | -5 | -2 | -79 | -48 | -13 | -19 | 0 | -33 | -13 | -17 |
| NUM | -26 | -0 | -25 | -23 | -7 | -1 | -19 | -78 | -11 | -2 | 0 | -34 | -10 | -9 |
| PRON | -6 | -1 | -7 | -1 | -3 | -8 | -2 | -24 | -73 | -4 | 0 | -42 | -3 | -5 |
| PROPN | -14 | -0 | -16 | -18 | -5 | -1 | -25 | -25 | -14 | -73 | 0 | -30 | -10 | -10 |
| PUNCT | -5 | -2 | -22 | -19 | -4 | -1 | -9 | -3 | -15 | -10 | -14 | -14 | -11 | -8 |
| SCONJ | -6 | -8 | -35 | -5 | -11 | -3 | -4 | -45 | -13 | -10 | 0 | -42 | -11 | -6 |
| VERB | -22 | -0 | -13 | -13 | -2 | -2 | -6 | -41 | -5 | -15 | 0 | -16 | -67 | -9 |

(row labels: English Neutralizer)

**Greek Target — English Neutralizer (right)**

| | ADJ | ADP | ADV | AUX | CCONJ | DET | NOUN | NUM | PART | PRON | PROPN | PUNCT | SCONJ | VERB | X | avg |
|---|---|---|---|---|---|---|---|---|---|---|---|---|---|---|---|---|
| ADJ | -59 | -1 | -19 | -1 | -3 | -0 | -7 | -15 | -8 | -12 | -28 | -0 | -10 | -0 | -20 | -7 |
| ADP | -3 | -23 | -9 | -2 | -3 | 0 | -1 | -2 | -2 | -2 | -10 | 0 | -13 | -0 | -24 | -2 |
| ADV | -5 | -1 | -75 | -1 | -3 | 0 | -2 | -6 | -4 | -5 | -6 | 0 | -10 | -1 | -22 | -3 |
| AUX | -3 | -1 | -12 | -56 | -2 | -0 | -1 | -5 | 0 | -3 | -0 | -0 | -11 | -18 | -8 | -5 |
| CCONJ | -0 | -1 | -10 | -3 | -61 | 0 | -3 | -4 | 0 | -1 | -8 | -0 | -4 | -1 | -15 | -2 |
| DET | -14 | -1 | -2 | -7 | -1 | -9 | 0 | -22 | 0 | -8 | -8 | -0 | -9 | -2 | -4 | -4 |
| NOUN | -13 | -0 | -16 | -6 | -3 | -0 | -69 | -20 | 0 | -14 | -33 | -2 | -2 | -3 | -25 | -16 |
| NUM | -5 | -1 | -7 | 0 | -1 | -0 | -19 | -69 | -4 | -3 | -12 | -0 | -3 | -4 | 0 | -6 |
| PART | -1 | -1 | -7 | -7 | -1 | 0 | 0 | -3 | 0 | -1 | 3 | -0 | 0 | -2 | -17 | -1 |
| PRON | -1 | 0 | -8 | -1 | 0 | 0 | -0 | -0 | 0 | -37 | -2 | 0 | -7 | 1 | 3 | -1 |
| PROPN | -5 | -0 | -9 | 0 | -2 | -0 | -11 | -8 | 1 | -6 | -22 | 0 | -1 | -2 | -37 | -4 |
| PUNCT | -1 | -0 | -11 | -3 | -4 | 0 | -9 | -17 | -14 | -6 | -10 | -46 | -10 | -1 | -57 | -9 |
| SCONJ | -4 | -3 | -63 | -1 | -2 | 0 | -4 | -9 | 1 | -4 | -4 | -0 | -9 | -0 | -36 | -3 |
| VERB | -8 | -0 | -8 | -6 | -1 | 0 | -6 | -8 | 1 | -7 | -10 | 0 | -2 | -34 | -8 | -5 |
| X | -10 | 0 | -6 | 0 | 0 | 0 | -51 | -23 | 1 | -3 | -23 | 0 | 1 | -0 | -24 | -12 |

(row labels: English Neutralizer)

Figure 12: Relative change in accuracy when cross-neutralizing Italian (left) and Greek (right) POS tag embeddings using English centroids, using XLM-R representations.

**English Target — Italian Neutralizer (left)**

| | ADJ | ADP | ADV | AUX | CCONJ | DET | NOUN | NUM | PRON | PUNCT | SCONJ | VERB | avg |
|---|---|---|---|---|---|---|---|---|---|---|---|---|---|
| ADJ | -63 | -3 | -25 | -13 | -6 | -3 | -11 | -19 | -4 | -1 | -33 | -13 | -10 |
| ADP | -4 | -61 | -14 | -3 | -2 | -2 | -0 | -14 | -3 | -0 | -47 | -2 | -6 |
| ADV | -15 | -1 | -88 | -5 | -2 | -1 | -6 | -18 | -1 | -0 | -7 | -1 | -6 |
| AUX | -9 | -6 | -7 | -50 | -1 | -0 | -5 | -9 | -1 | -0 | -8 | -42 | -7 |
| CCONJ | -2 | -2 | -25 | -2 | -78 | -5 | -4 | -5 | -2 | -0 | -16 | -0 | -4 |
| DET | -2 | -2 | -4 | -2 | 0 | -9 | -1 | -6 | -7 | -0 | -18 | -2 | -2 |
| NOUN | -21 | -4 | -25 | -14 | -9 | -5 | -79 | -30 | -13 | -1 | -46 | -14 | -20 |
| NUM | -12 | -7 | -23 | -5 | -3 | -5 | -30 | -61 | -9 | -0 | -52 | -10 | -13 |
| PRON | -2 | -1 | -6 | -2 | -2 | -1 | -7 | -2 | -22 | -0 | -17 | -5 | -3 |
| PUNCT | -13 | -3 | -35 | -6 | -12 | -6 | -23 | -14 | -3 | -100 | -61 | -8 | -22 |
| SCONJ | -2 | -2 | -32 | -3 | -3 | -0 | -5 | -7 | -1 | -0 | -61 | -1 | -3 |
| VERB | -20 | -3 | -15 | -19 | -7 | -3 | -6 | -11 | -3 | -1 | -28 | -81 | -11 |

(row labels: Italian Neutralizer)

**Greek Target — Italian Neutralizer (right)**

| | ADJ | ADP | ADV | AUX | CCONJ | DET | NOUN | NUM | PRON | PROPN | PUNCT | SCONJ | VERB | X | avg |
|---|---|---|---|---|---|---|---|---|---|---|---|---|---|---|---|
| ADJ | -48 | -0 | -18 | -1 | -3 | -0 | -12 | -14 | -7 | -36 | -0 | -13 | 0 | -25 | -7 |
| ADP | -6 | -51 | -12 | -1 | -1 | 0 | -0 | -8 | -4 | -12 | 0 | -12 | -0 | -36 | -5 |
| ADV | -11 | -1 | -85 | 0 | -2 | 0 | -2 | -5 | -7 | -21 | 0 | -7 | 0 | -43 | -5 |
| AUX | -5 | -3 | -6 | -57 | -1 | -1 | -1 | -8 | -4 | -0 | 0 | -9 | -34 | -17 | -6 |
| CCONJ | -1 | -0 | -20 | -3 | -56 | -0 | -5 | -12 | -1 | -11 | -0 | -9 | -1 | -11 | -3 |
| DET | -20 | -2 | -5 | -12 | -3 | -49 | -3 | -22 | -14 | -41 | 0 | -9 | -4 | -17 | -11 |
| NOUN | -12 | 0 | -10 | -1 | -0 | -0 | -76 | -17 | -16 | -43 | -2 | -2 | -3 | -32 | -17 |
| NUM | -10 | -1 | -8 | -3 | -0 | -0 | -53 | -84 | -2 | -29 | -1 | -6 | -8 | -11 | -15 |
| PRON | -5 | 0 | -5 | -1 | 0 | -0 | -2 | -3 | -34 | -17 | 0 | -12 | 1 | -19 | -2 |
| PROPN | -4 | 0 | -5 | -1 | -0 | -0 | -10 | -6 | -2 | -16 | 0 | -3 | -3 | -42 | -3 |
| PUNCT | -6 | -34 | -74 | -33 | -13 | -0 | -14 | -26 | -7 | -14 | -100 | -6 | -10 | -73 | -23 |
| SCONJ | -3 | -1 | -68 | -1 | -3 | -0 | -4 | -9 | -5 | -10 | -0 | -17 | -0 | -48 | -4 |
| VERB | -17 | -0 | -8 | -12 | -2 | 0 | -5 | -12 | -6 | -26 | 0 | -3 | -42 | -25 | -6 |
| X | -8 | -0 | -5 | 0 | -1 | 0 | -16 | -9 | -7 | -27 | 0 | -2 | 0 | -93 | -5 |

(row labels: Italian Neutralizer)

Figure 13: Relative change in accuracy when cross-neutralizing English (left) and Greek (right) POS tag embeddings using Italian centroids, using XLM-R representations.

Figure 14: Relative change in accuracy when cross-neutralizing English (left) and Italian (right) POS tag embeddings using Greek centroids, using XLM-R representations.

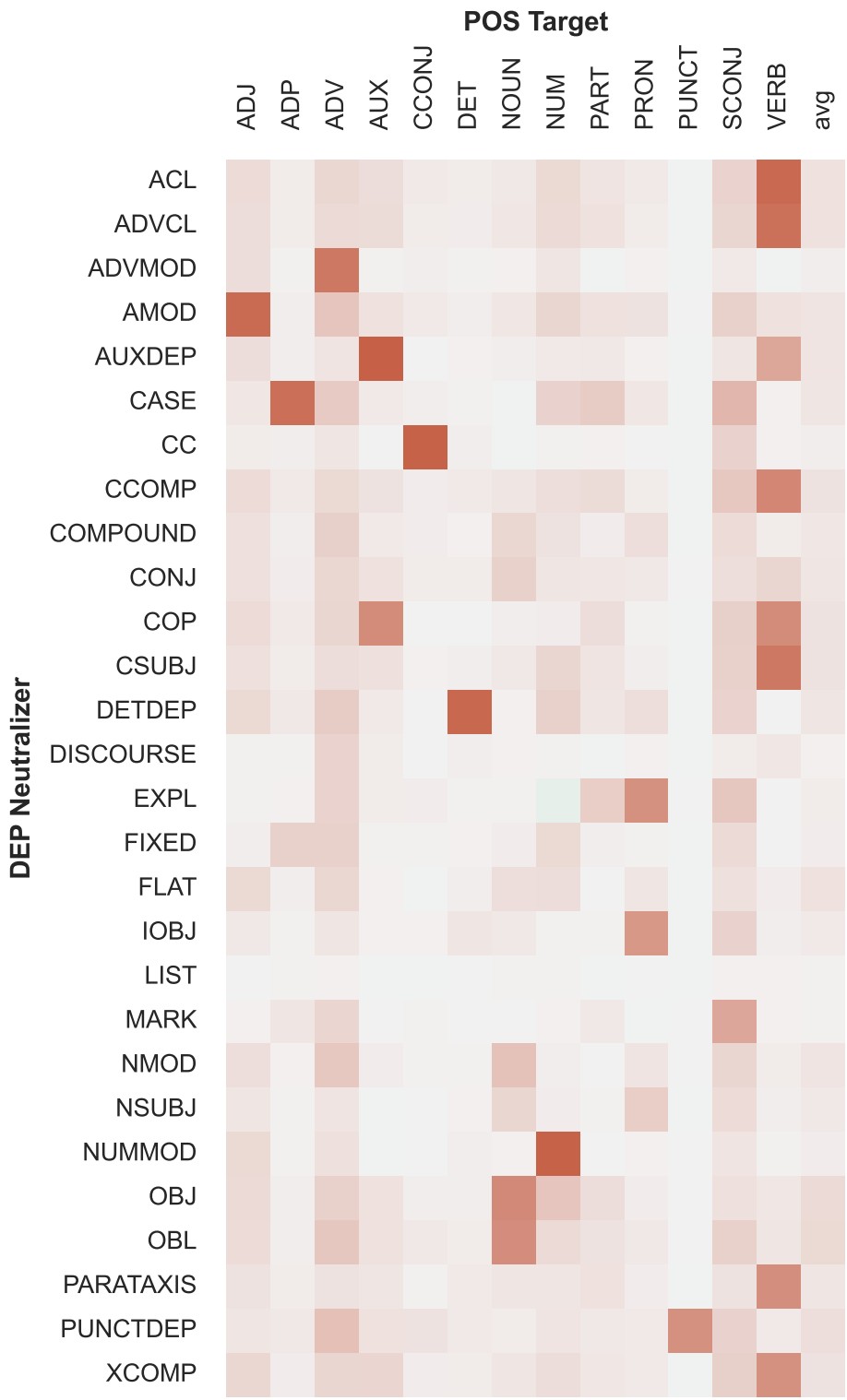

Figure 15: Relative change in accuracy when cross-neutralizing RoBERTa POS embeddings using RoBERTa DEP centroids.

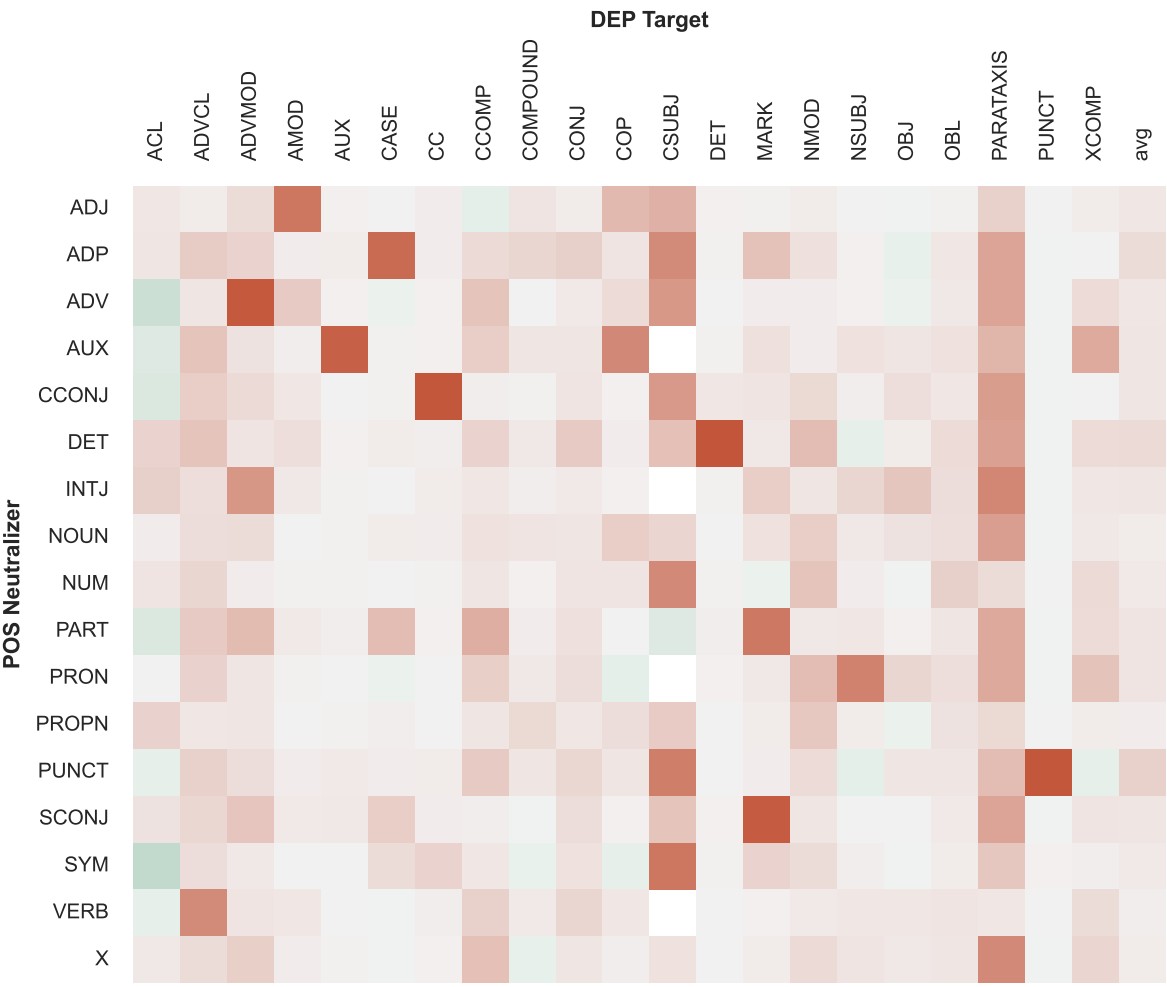

Figure 16: Relative change in accuracy when cross-neutralizing RoBERTa DEP embeddings using RoBERTa POS centroids.

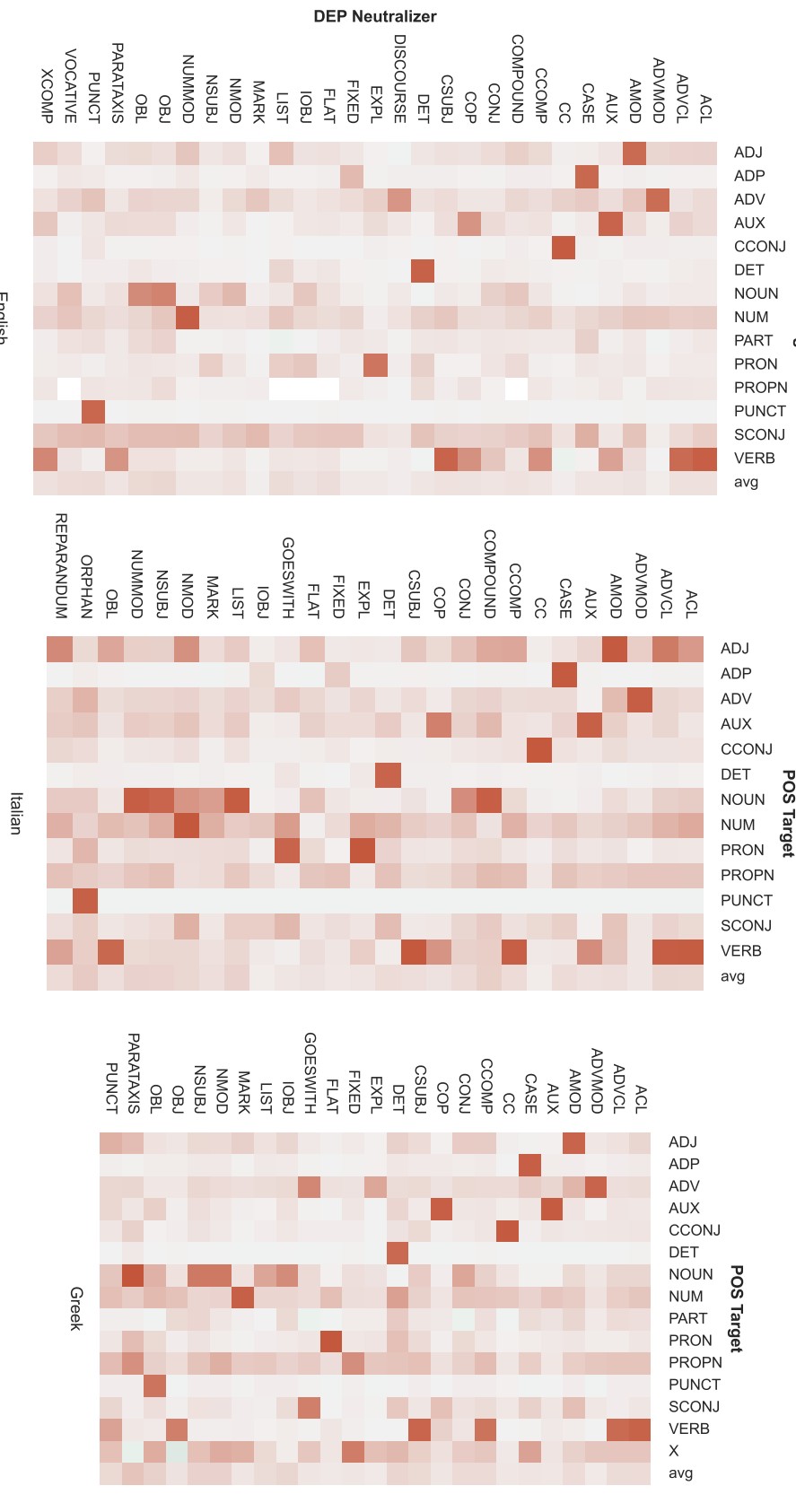

Figure 17: Relative change in accuracy when cross-neutralizing XLM-R POS embeddings using XLM-R DEP centroids. In English, Italian and Greek. Some languages have less columns or rows than others because the particular category for the missing row/column is not in their dataset.

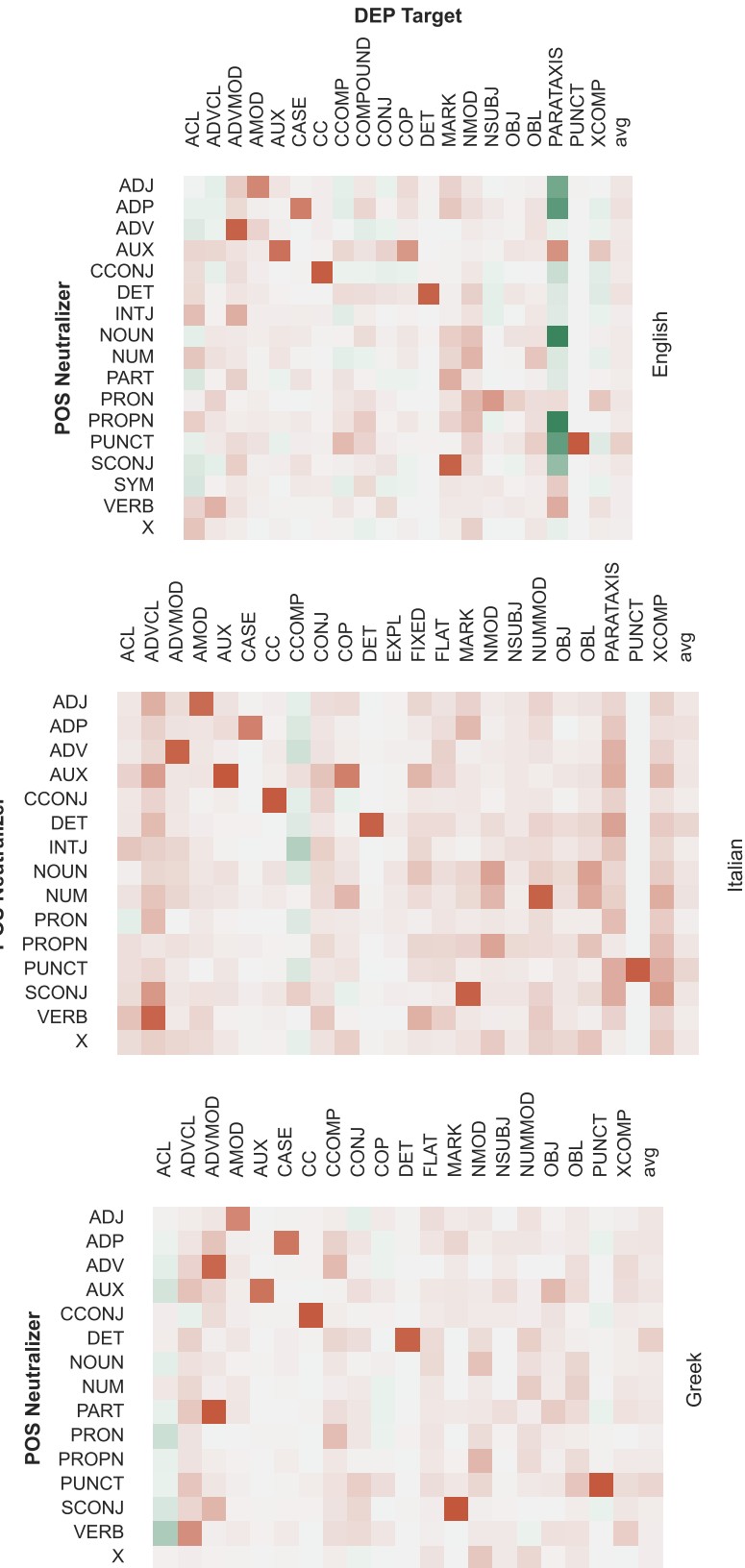

Figure 18: Relative change in accuracy when cross-neutralizing XLM-R DEP embeddings using XLM-R POS centroids. In English, Italian and Greek. Some languages have less columns or rows than others because the particular category for the missing row/column is not in their dataset.