# OpenReview forum: "Probing LLMs for Joint Encoding of Linguistic Categories"
_EMNLP/2023/Conference — EMNLP 2023 Findings_

### Official Review · Reviewer_ZArK · 2023-08-02

**Soundness:** 4

**Excitement:**

3: Ambivalent: It has merits (e.g., it reports state-of-the-art results, the idea is nice), but there are key weaknesses (e.g., it describes incremental work), and it can significantly benefit from another round of revision. However, I won't object to accepting it if my co-reviewers champion it.

**Paper Topic And Main Contributions:**

This paper investigates the ability of Large Language Models (LLMs) to joint encoding linguistic categories. In particular, it builds on previous research suggesting that LLMs exhibit a linguistic hierarchy in their layers, with lower layers being more suitable for syntactic tasks and higher layers for semantic processing. The authors propose a framework for testing the joint encoding of linguistic categories in LLMs, focusing on syntax and, in particular, on POS tagging and DEP labelling. They find evidence of joint encoding at both the same level, such as related part-of-speech (POS) classes, and different levels, such as POS classes and related syntactic dependency relations. Furthermore, their cross-lingual experiments demonstrate that these patterns hold across languages in multilingual LLMs.

**Reasons To Accept:**

-- The paper is well-written and interesting to read.

-- The approach devised to understand how LLMs jointly encode linguistic properties is very interesting. By investigating the joint encoding of linguistic categories, the paper sheds light on the model's ability to learn diverse properties simultaneously. This aligns with the existing line of research and contributes to our understanding of the capabilities of these language models.

-- The discussion of the results is insightful and adds value to the findings. The paper not only presents empirical evidence but also provides an analysis and interpretation of the observed patterns, giving the reader a deeper understanding of the implications of the research.

**Reasons To Reject:**

-- Despite the interesting approach and the insightful discussion of the results, the contribution of the paper appears somewhat limited. The research could have been better presented as a short paper. Additionally, more experiments would have been beneficial for better appreciating the findings, such as testing different models at various sizes. Moreover, the authors experimented only using representations from layers 1, 3, 6, 9 and 12 and subsequently selecting the best configuration for each task, language and model. It would have been also very interesting to see how the ability of the model to jointly encode linguistic properites changes across layers.

-- The author tested a two-layer MLP model as probing model. However, in the last few years, there has been a long debate about the validity of probing models and how to choose the best model configuration (e.g. very simple linear models vs. more complex NN models). I think that this issue should have been taken into account to give more validity to the findings of this work.

-- The introduction suggests that understanding how LLMs represent linguistic phenomena is crucial for developing more effective NLP methods. However, the paper lacks a discussion on the possible implications of the study's findings. Although I think that not all research needs to have direct application purposes (especially in the context of interpretability studies), it would have been interesting to see a discussion on the potential implications and applications of these results for building more efficient NLP systems, as mentioned in the introduction. Such insights would add further value to the research and its relevance to the NLP community.

**Reproducibility:**

4: Could mostly reproduce the results, but there may be some variation because of sample variance or minor variations in their interpretation of the protocol or method.

**Reviewer Confidence:**

4: Quite sure. I tried to check the important points carefully. It's unlikely, though conceivable, that I missed something that should affect my ratings.

---

> ### Author Rebuttal · Authors · 2023-08-29
>
> Thank you very much for your time and helpful feedback! We are glad to hear that you enjoyed reading our paper, and found the approach interesting and analyses insightful.
>
> We agree that experiments on the effect of model size would certainly add value, particularly given recent research on scaling laws. Following your suggestion, we repeated our proposed methodology for POS tag cross-neutralization on RoBERTa-large. Compared to RoBERTa-base, We notice that similar patterns occur across model sizes (e.g. NOUN neutralizing ADV). The effect appears to be generally less pronounced for larger models, although we also did find instances where the drop in accuracy is the same or higher. We repeated the same experiment with XLM-R-base, XLM-R-large and XLM-R-XL and observed the same patterns and trends. We will include these results in the camera-ready version.
>
> As for our treatment of the transformer layers, we devoted attention to this in the work because we expected potentially significant differences in how joint encoding occurs depending on the layer given the work of Tenney et al. (2019). However, we later found this not to be the case, hence the lack of results in this theme. This is perhaps due to the relatively low-level nature of the tasks considered (in terms of the linguistic hierarchy), and may change for higher-level NLP tasks (e.g. more complex NLU tasks). We note that this can be partially inferred from Fig. 7, in the appendix, where the self-neutralizing accuracy drop is rather significant (between 80% and 95%) regardless of layer configuration, and no strong trend can be identified. We do realize that this is not immediately clear in the main text, and we will make sure to further elaborate on this in the camera-ready version.
>
> As for our choice of probing classifier, we agree that considerations are speaking for and against both the linear and the non-linear probes. To further validate the results of our probes, based on the suggestions of the reviewers, we ran a selectivity analysis on the POS tagging task, as proposed by Hewitt and Liang (2019). We train our probing classifier on RoBERTa-base representations on the true POS tagging task, and then train a new checkpoint for the same number of steps on the control task. Under this setup, we find that our probing classifier achieves an average accuracy of 96% on the POS tagging task and 63% on the control task, giving a selectivity of 33%. We believe this should be sufficient to indicate our results are indeed valid. We will make sure to include this baseline in the camera-ready version.
>
> We believe that our method and findings are important as they shed light on the ways in which LLMs generalize linguistic information. In this paper, we tested specifically for joint encoding of different syntactic categories that rely on the same linguistic concept (e.g. an “amod” dependency implies some relationship with the adjective and the noun POS tags). However, the approach can be extended to test for joint encoding of classes/labels/categories in other tasks that can be expected to share information. This in turn would inform us on whether an LM indeed captures this shared information between different tasks or if does not. Knowing this can help us develop better models in many application scenarios: for instance, in a multitask learning setting (where negative interference between parameter updates from different tasks is known to hamper performance) or a lifelong learning setting (where learning a new task often leads to (catastrophic) forgetting of the previously learned ones). We will add a discussion of this in the final version of the paper.
>
> We would like to end with a quote from John Hewitt: "Gaining insights into the natures of NLP’s unsupervised representations may help us to understand why our models succeed and fail, what they’ve learned, and what we yet need to teach them”. This is something we very much agree on.
>
> ## References
>
> 1. Hewitt J, Liang P. Designing and Interpreting Probes with Control Tasks. In: Proceedings of the 2019 Conference on Empirical Methods in Natural Language Processing and the 9th International Joint Conference on Natural Language Processing (EMNLP-IJCNLP) [Internet]. Hong Kong, China: Association for Computational Linguistics; 2019
> 2. Tenney I, Das D, Pavlick E. BERT Rediscovers the Classical NLP Pipeline. In: Proceedings of the 57th Annual Meeting of the Association for Computational Linguistics [Internet]. Florence, Italy: Association for Computational Linguistics; 2019. p. 4593–601.

---

### Official Review · Reviewer_HBxT · 2023-08-07

**Soundness:** 4

**Excitement:**

3: Ambivalent: It has merits (e.g., it reports state-of-the-art results, the idea is nice), but there are key weaknesses (e.g., it describes incremental work), and it can significantly benefit from another round of revision. However, I won't object to accepting it if my co-reviewers champion it.

**Missing References:**

- Amensic probing / INLP: https://aclanthology.org/2020.acl-main.647/ , https://aclanthology.org/2021.tacl-1.10
- Crosslingual morphosyntactic probing, with an eye to joint-encoding-like phenomena: https://aclanthology.org/2021.findings-emnlp.382/ , https://aclanthology.org/2022.naacl-main.114/

**Paper Topic And Main Contributions:**

This paper explores "joint encoding" of linguistic properties in both mono and multilingual language models.  Two properties are "jointly encoded" if they are encoded in similar portions of the vector space; the authors measure this by taking the centroid of the representations for one property (class, e.g. POS tag) and subtracting it from the representations and seeing if it effects performance on another category (e.g. another POS, dependency labeling, or same tag in another language).  The results are suggestive: many properties do exhibit joint encoding, and there are also similarities in the effects across languages within one multilingual model.  The paper presents interesting experiments and results, but could use more motivation for the approach and more detailed comparison with other methods.

AFTER REBUTTAL EDIT: given the news about the baselines, and the enriched discussion of INLP, I am happy to upgrade my score from 3 to 4.

**Questions For The Authors:**

- Why did you choose to use a two-layer MLP for your probes (as opposed to a simple linear model)?  Did you also look at selectivity (performance on a control task) to make sure that the probing model itself is not too powerful?
- Is there more theoretical justification for subtracting-the-centroid as removing the information?  I'd be curious to hear more about a comparison with amnesic probing / iterative nullspace projection, which removes the linear information of an attribute via projection onto the nullspace.  Would that be an alternative to your method?
- Why dependency label prediction as a task, and not as well whether or not there is a dependence relation?
- Why were English, Italian, and Greek the only three languages studied?  How were they chosen?

**Reasons To Accept:**

- Interesting question about the similarity of representations across tasks and across languages.
- Analyzes both mono- and multi-lingual models.

**Reasons To Reject:**

- Lack of baselines, both for the probing models (e.g. selectivity), and for the subtraction method.  It would be good to subtract a random vector, or the centroid of all word representations (or some such) to see how much worse the given subtractions are.
- Could benefit from more thorough discussion of alternative attribute-removal methods (see questions / references) and other studies looking at crosslingual joint encoding.
- Some analyses are merely speculative, e.g. for across-POS differences.

**Reproducibility:**

4: Could mostly reproduce the results, but there may be some variation because of sample variance or minor variations in their interpretation of the protocol or method.

**Reviewer Confidence:**

3: Pretty sure, but there's a chance I missed something. Although I have a good feel for this area in general, I did not carefully check the paper's details, e.g., the math, experimental design, or novelty.

---

> ### Author Rebuttal · Authors · 2023-08-29
>
> Thank you very much for your time and helpful suggestions!
>
> ## Baselines
>
> As you suggested, we tested two additional baselines to provide a more comprehensive evaluation. We ran a selectivity baseline on the POS tagging task which is easier to memorize, following Hewitt and Liang (2019). We train our probing classifier on RoBERTa-base representations on the true POS tagging task, and then train a new checkpoint for the same number of steps on the control task. Under this setup, we find that our probing classifier achieves an average accuracy of 96% on the POS tagging task and 63 % on the control task, giving a selectivity of 33%. We believe this should be sufficient to indicate our results are indeed valid. We will make sure to include this baseline in the camera-ready version.
>
> In the second baseline, we subtracted random vectors (rather than centroids) from the model representations. We found that this left the performance relatively unchanged, with no evidence of the systematic drops observed when subtracting centroids. Both additional experiments strengthen our results, and we are very grateful to you and your fellow reviewers for the suggestions! We will make sure to include both of these experiments in the camera-ready version of the paper.
>
> ## Other Questions
>
> We’d like to thank you for pointing us to the related work on INLP and amnesic probing which would pose an interesting alternative to our proposed method. Indeed these would appear to be valid substitutes for subtraction. We agree that our paper would benefit from a more thorough comparison between INLP and our method for completeness. We will make sure to include this in the camera-ready version of the paper.
>
> Regarding your first question, we decided to use a two-layer MLP following Choenni and Shutova (2022), Şahin et al. (2020) and Tenney et al. (2019) who do the same. Note that with “two-layer MLP” we mean that our classifier consists of two linear layers with a non-linear activation in between. We will clarify this in the camera-ready version.
>
> As for your third question, we chose to focus on dependency label prediction rather than dependency parsing mainly for simplicity. Because we were mainly concerned with identifying representations tied to specific dependency labels, we envisioned the labeling task as sufficient for this. Full dependency parsing, including identifying dependency arcs, may have introduced an additional capability requirement to our probing setup and would have made it more difficult to disentangle the task of assigning dependency types from the task of identifying the dependency arcs.
>
> With regard to the final question, these three specific languages were chosen to facilitate our analysis and interpretation of the results, as at least one author was familiar with each.
>
>
> ## References
>
>  1. Şahin GG, Vania C, Kuznetsov I, Gurevych I. LINSPECTOR: Multilingual Probing Tasks for Word Representations. Computational Linguistics. 2020 Jun;46(2):335–85.
> 2. Choenni R, Shutova E. Investigating Language Relationships in Multilingual Sentence Encoders Through the Lens of Linguistic Typology. Computational Linguistics. 2022 Sep;48(3):635–72.
> 3. Hewitt J, Liang P. Designing and Interpreting Probes with Control Tasks. In: Proceedings of the 2019 Conference on Empirical Methods in Natural Language Processing and the 9th International Joint Conference on Natural Language Processing (EMNLP-IJCNLP) [Internet]. Hong Kong, China: Association for Computational Linguistics; 2019
> 4. Tenney I, Xia P, Chen B, Wang A, Poliak A, McCoy RT, et al. What do you learn from context? Probing for sentence structure in contextualized word representations. In 2019.

---

### Official Review · Reviewer_neqX · 2023-08-10

**Soundness:** 3

**Excitement:**

3: Ambivalent: It has merits (e.g., it reports state-of-the-art results, the idea is nice), but there are key weaknesses (e.g., it describes incremental work), and it can significantly benefit from another round of revision. However, I won't object to accepting it if my co-reviewers champion it.

**Missing References:**

Some previous work on how syntactic information is stored in Transformer based models and how it evolves within the network are missing from the background section. Specifically:

- @misc{dalvi2022discovering,
      title={Discovering Latent Concepts Learned in BERT},
      author={Fahim Dalvi and Abdul Rafae Khan and Firoj Alam and Nadir Durrani and Jia Xu and Hassan Sajjad},
      year={2022},
      eprint={2205.07237},
      archivePrefix={arXiv},
      primaryClass={cs.CL}
}





**Paper Topic And Main Contributions:**

The authors proposed a framework for testing joint encodings of linguistic categories in Large Language Models. They extended the idea of "cross-neutrilization" to study how information is encoded across different linguistic categories in two syntactic tasks (Part-of-Speech tagging and dependency labeling). The contribution of the paper are summarized below:
- Proposed a framework for analysis of joint encodings in language models.
- Provided evidence for joint encoding between different POS tags.
- Provided evidence for joint encoding across languages.
- Provided evidence for shared knowledge across tasks.

**Questions For The Authors:**

Question A: You mention that you finetune a probing classifier. What was the probe trained on in the first place?

Question B: Why didn’t you report the selectivity of the trained probe?

Question C: You mentioned in the introduction that one of the research questions that the paper addresses are “how syntactic categories” are stored within the models. Why didn’t you address this question?


**Reasons To Accept:**

- Approach suggested helps in understanding how linguistic categories are encoded within language models, how they are shared between tasks, and how they are shared between languages.


**Reasons To Reject:**

- Only reported classification accuracy of the probes trained without reporting selectivity. The probe trained might be memorizing the data rather than learning to predict the linguistic categories.

**Reproducibility:**

4: Could mostly reproduce the results, but there may be some variation because of sample variance or minor variations in their interpretation of the protocol or method.

**Reviewer Confidence:**

4: Quite sure. I tried to check the important points carefully. It's unlikely, though conceivable, that I missed something that should affect my ratings.

**Typos Grammar Style And Presentation Improvements:**

- Text in figure 1 is small. Please consider increasing font size

---

> ### Author Rebuttal · Authors · 2023-08-29
>
> Thank you very much for your time and helpful feedback!
>
> Regarding question A, we apologize for the unclear language here. The probing classifier itself is not pretrained on anything, and rather is randomly initialized. So in a certain sense saying we fine-tune the probing classifier is incorrect, and we understand your confusion. Instead, we should have said we _train_ the probing classifier on the representations incoming from the frozen pre-trained encoder. We will make sure to correct this in the camera-ready version.
>
> Regarding the selectivity baseline: Thank you for this suggestion! Following your advice, we now ran a selectivity baseline on the POS tagging task which is easier to memorize, following Hewitt and Liang (2019). We train our probing classifier on RoBERTa-base representations on the true POS tagging task, and then train a new checkpoint for the same number of steps on the control task. Under this setup, we find that our probing classifier achieves an average accuracy of 96% on the POS tagging task and 63% on the control task, giving a selectivity of 33%. We believe this should be sufficient to indicate our results are indeed valid. We will make sure to include this baseline in the camera-ready version.
>
> Regarding question C, the “syntactic categories” we refer to in the introduction are precisely the POS tags and dependency labels we study. For instance, we may refer to the NOUN POS tag as a syntactic category. We envision our results, showing that many of these categories are jointly encoded in different ways (e.g. across tasks and/or languages), shed some light on how these categories are encoded in the models, one of the questions we pose in the introduction.
> We will make sure to clarify this in the camera-ready version.
>
> Last but not least, thank you for the feedback on the figure. We will make sure to increase the font size.
>
> ## References
>
> 1. Hewitt J, Liang P. Designing and Interpreting Probes with Control Tasks. In: Proceedings of the 2019 Conference on Empirical Methods in Natural Language Processing and the 9th International Joint Conference on Natural Language Processing (EMNLP-IJCNLP) [Internet]. Hong Kong, China: Association for Computational Linguistics; 2019

---

### Meta-Review · Area_Chair_Mj34 · 2023-09-15

**Recommendation:** 4

**Metareview:**

This paper presents a framework for analyzing joint encodings in LMs, and evidence for the existing of these for certain labels and tasks. The reviewers identified missing baselines as well as additional experiments (e.g. different model sizes); but these have all been provided by the authors in the rebuttal. It is well written, and has enough significance (especially after the rebuttal). The leftover cons would be the question of validity of a 2-layer probe (this is a larger discussion within probing), the lack of discussion of implications, and that some analyses are merely speculative. I would say the first one is beyond the scope of this paper, the discussion of implications could be added, and speculative analyses could be seen as interesting avenues for future work.

---

### Decision · Program_Chairs · 2023-10-07

**Decision:**

Accept-Findings

**Comment:**

This paper presents a framework for analyzing joint encodings in LMs, and evidence for the existing of these for certain labels and tasks. The reviewers identified missing baselines as well as additional experiments (e.g. different model sizes); but these have all been provided by the authors in the rebuttal. It is well written, and has enough significance (especially after the rebuttal). The leftover cons would be the question of validity of a 2-layer probe (this is a larger discussion within probing), the lack of discussion of implications, and that some analyses are merely speculative. I would say the first one is beyond the scope of this paper, the discussion of implications could be added, and speculative analyses could be seen as interesting avenues for future work.